# Multi-Task Reinforcement Learning with Soft Modularization

**Ruihan Yang**[1]    **Huazhe Xu**[2]    **Yi Wu**[3,4]    **Xiaolong Wang**[1]

[1]UC San Diego        [2] UC Berkeley        [3] IIIS, Tsinghua        [4] Shanghai Qi Zhi Institute

## Abstract

Multi-task learning is a very challenging problem in reinforcement learning. While training multiple tasks jointly allow the policies to share parameters across different tasks, the optimization problem becomes non-trivial: It remains unclear what parameters in the network should be reused across tasks, and how the gradients from different tasks may interfere with each other. Thus, instead of naively sharing parameters across tasks, we introduce an explicit modularization technique on policy representation to alleviate this optimization issue. Given a base policy network, we design a routing network which estimates different routing strategies to reconfigure the base network for each task. Instead of directly selecting routes for each task, our task-specific policy uses a method called *soft modularization* to softly combine all the possible routes, which makes it suitable for sequential tasks. We experiment with various robotics manipulation tasks in simulation and show our method improves both sample efficiency and performance over strong baselines by a large margin. Our project page with code is at https://rchalyang.github.io/SoftModule/.

## 1 Introduction

Deep Reinforcement Learning (RL) has recently demonstrated extraordinary capabilities in multiple domains, including playing games [21] and robotic control and manipulation [18, 16]. Despite its successful applications, Deep RL still requires a large amount of data for training complex tasks. On the other hand, while the current deep RL methods can learn individual policies for specific tasks such as robot grasping and pushing, it remains very challenging to train a single network that generalizes across all possible robotic manipulation tasks.

In this paper, we study multi-task RL as one step forward towards skill sharing across diverse tasks and ultimately building robots that can generalize. Training deep networks with multiple tasks jointly, agents can learn to share and re-use components across different tasks, which further leads to improved sample efficiency. This is particularly important when we want to adopt RL algorithms in real-world applications. Multi-task learning also provides a natural curriculum since learning easier tasks can be beneficial for learning of more challenging tasks with shared parameters [25].

However, multi-task RL remains a hard problem. It becomes even more challenging when the number of tasks increases. For instance, it has been shown by [43] that training with diverse robot manipulation tasks jointly with a sharing network backbone and multiple task-specific heads for actions hurt the final performance comparing to independent training in each task. One major reason is that multi-task learning introduces optimization difficulties: It is unclear how the tasks will affect each other when trained jointly, and optimizing some tasks can bring negative impacts on the others [37].

For tackling this problem, compositional models with multiple modules were introduced [1, 9]. For example, researchers proposed to train modular sub-policies and task-specific high-level policies jointly in a Hierarchical Reinforcement Learning (HRL) framework [1]. The sub-policies can be shared and selected by different high-level policies with a learned policy composition function.

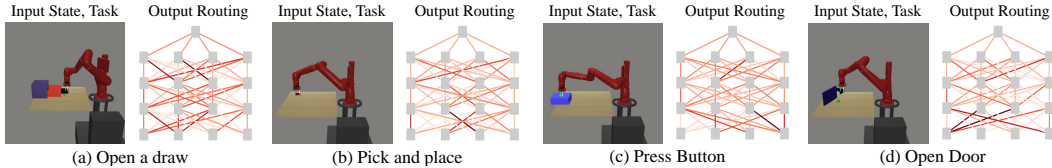

| Input State, Task | Output Routing | Input State, Task | Output Routing | Input State, Task | Output Routing | Input State, Task | Output Routing |

(a) Open a draw           (b) Pick and place          (c) Press Button          (d) Open Door

Figure 1: Our multi-task policy network with soft modularization. Given different tasks, our network generate different soft combination of network modules. Gray squares represent network modules and red lines represent the connection between modules (Darker red indicates larger weight).

However, HRL introduces an optimization challenge on jointly training sub-policies and high-level task-specific policies while training sub-policies separately often require predefined subtasks or some sophisticated way to discover subgoals, which are typically infeasible for real-world applications.

In this paper, instead of designing individual modules explicitly for each sub-policy, we propose a soft modularization method that generates soft combinations of different modules for different tasks automatically without explicitly specifying the policy structure. This approach consists of two networks: a base policy network and a routing network. The base policy network, which is composed of multiple modules, takes the state as input and outputs an action for the task. The routing network takes a task embedding and the current state as input and estimates the routing strategy.

Given a task, the modules in the base policy network will be reconfigured by the routing network. This is visualized in Figure 1. Furthermore, instead of taking hard assignments on modules, which is hard to optimize in sequential tasks, our routing network outputs a probability distribution over module assignments for each task. A task-specific base network can be viewed as a weighted combination of the shared modules according to the probability distribution. We benefit from this design to directly back-prop through the routing weights and train both networks jointly over multiple tasks. The advantage is that we can modularize the networks according to tasks without specifying policy hierarchies explicitly (e.g., HRL). The role of each module automatically emerged after training and the routing network determines which modules should be used more for different tasks.

We perform experiments in Meta-World [43], which contains 50 robotic manipulation tasks. With soft modularization, we achieve significant improvements in both sample efficiency and final performance over previous state-of-the-art multi-task policies. For example, we almost double the manipulation success rate for learning with 50 tasks compared to the multi-task baselines. Our approach utilizes far less training data compared to training individual policies for each task while achieving learned policy that is able to perform closely to the individually trained policies. This shows that enforcing soft modularization can improve the generalization across different tasks in RL.

## 2 Related Work

**Multi-task learning.** Multi-task learning [3] is one of the core machine learning problems. Researchers have shown learning with multiple objectives can make different tasks benefit from each other in robotics and RL [41, 24, 25, 28, 11, 34]. While sharing parameters across tasks can intuitively improve data efficiency, gradients from different tasks can interfere negatively with each other. One way to avoid it is to use policy distillation [22, 31, 37, 37]. However, these approaches still require separate networks for different policies and an extra distillation stage. Researchers also propose to explicitly model the similarity between gradients from different tasks [44, 4, 15, 19, 35, 6, 42, 12]. For example, it is proposed in [4] to normalize the gradients from different tasks for balancing multi-task losses. Besides adjusting the losses, a recent work [42] proposes to directly reduce the gradient conflicts by gradient projection. However, optimization relying on the gradient similarity is usually unstable, especially when there is a large gradient variance within each task itself.

**Compositional learning and modularization.** Instead of directly enforcing the gradients to align, a natural way to avoid the conflicts of the gradient is using compositional models. By utilizing different modules across different tasks, it reduces the interference of gradients on each module and allows better generalization [36, 5, 1, 32, 27, 23, 9, 33, 8]. For example, the policy is decomposed to task-specific and robot-specific modules in [5], and the policy is able to solve unseen tasks by re-combining the pre-defined modules. However, the pre-defining modules and manual specification of the combination are not scalable. Instead of defining and pre-training the modules or sub-policies, our approach utilizes soft combinations over modules, which allows fully end-to-end training.

There are also related works learning a routing module in the supervised tasks. Rosenbaum et. al [30, 29] proposes to use RL to learn a routing policy, which can be considered as a *hard* version of our soft modularization method. This "hard modularization" approach is infeasible for RL settings because joint training a multitask control policy and a routing policy suffers from exponentially higher variance in policy gradient due to the temporal nature in RL and leads to severe training instability. For RL, "hard modularization" approach would further introduce high variance along with the exploration in the environment. Whereas, our soft version doesn't introduce additional variance, significantly stabilizes RL training, and produces much improved empirical performances. In addition, the following works inspire our work in different ways: Purushwalkam et. al [26] consider zero-shot compositional learning in vision; Wang et.al [40] consider weight generation for multi-task learning; Li et.al [17] alleviate the scale variance in semantic representation with dynamic routing.

**Mixture of experts.** Our method is also related to works on mixture of experts [14, 7, 13, 36, 2, 20]. For example, it is proposed in Satinder et. al [36] to train a gating function to select different Q functions (experts) for different tasks. Instead of performing one-time selection among the individual expert (which is usually pre-defined), the modules we propose are organized in multiple layers in our base policy network, with multiple layers of selection guided by the routing network. While each module alone is not functioning as a policy, this increases the flexibility of sharing the modules across different tasks. At the same time, it reduces the mutual interference between the modules because the modules are only connected via the routing network.

## 3 Background

We consider a finite horizon Markov decision processe (MDP) for each task $\mathcal{T}$ and there are $M$ tasks in total, which can be represented by $(S, A, P, R, H, \gamma)$, where the state $s \in S$ and action $a \in A$ are continuous. $P(s_{t+1}|s_t, a_t)$ represents the stochastic transition dynamics. $R(s_t, a_t)$ represents the reward function. $H$ is the horizon and $\gamma$ is the discount factor. We use $\pi_\phi(a_t|s_t)$ to represent the policy parameterized by $\phi$ and the goal is to learn a policy maximizing the expected return. In multi-task RL, tasks are sampled from a distribution $p(\mathcal{T})$, and different tasks have different MDPs.

### 3.1 Reinforcement Learning with Soft Actor-Critic

In this paper, we train policy with Soft Actor-Critic (SAC) [10]. SAC is an off-policy actor-critic deep reinforcement learning approach, where actor aims to succeed at the task as well as act as randomly as possible. We consider the parameterized soft Q-function is $Q_\theta(s_t, a_t)$ where $Q$ is parameterized by $\theta$. There are three types of parameters to optimize in SAC: The policy parameters $\phi$, the parameters of Q-function $\theta$ and a temperature $\alpha$. The objective of policy optimization is:

$$J_\pi(\phi) = \mathbb{E}_{s_t \sim \mathcal{D}} \left[ \mathbb{E}_{a_t \sim \pi_\phi} [\alpha \log \pi_\phi(a_t|s_t) - Q_\theta(s_t, a_t)] \right], \tag{1}$$

where $\alpha$ is a learnable temperature served as an entropy penalty coefficient. It can be learned to maintain the entropy level of the policy, using:

$$J(\alpha) = \mathbb{E}a_t \sim \pi_\phi \left[ -\alpha \log \pi_\phi(a_t|s_t) - \alpha \bar{\mathcal{H}} \right], \tag{2}$$

where $\bar{\mathcal{H}}$ is a desired minimum expected entropy. If $\log \pi_t(a_t|s_t)$ is optimized to increase its value, and the entropy is becoming smaller, $\alpha$ will be adjusted to increase in the process.

### 3.2 Multi-task Reinforcement Learning

We extend SAC from single task to multi-task by learning a single, task-conditioned policy $\pi(a|s, z)$, where $z$ represents a task embedding. We optimize the policy to maximize the average expected return across all tasks sampled from $p(\mathcal{T})$. The objective of policy optimization is,

$$J_\pi(\phi) = \mathbb{E}_{\mathcal{T} \sim p(\mathcal{T})} \left[ J_{\pi, \mathcal{T}}(\phi) \right], \tag{3}$$

where $J_{\pi, \mathcal{T}}(\phi)$ is adopted directly from Eq. 1 with task $\mathcal{T}$. Similarly for Q-function, the objective is:

$$J_Q(\theta) = \mathbb{E}_{\mathcal{T} \sim p(\mathcal{T})} \left[ J_{Q, \mathcal{T}}(\theta) \right]. \tag{4}$$

## 4 Method

We propose to perform multi-task reinforcement learning using a single base policy network with multiple modules. As visualized in Figure 2, instead of finding discrete routing paths to connect the modules for different tasks, we perform soft modularization: we utilize another routing network (right side of Figure 2) which takes the task identity embedding and observed state as inputs and outputs the probabilities to weight the modules in a soft manner.

With soft modularization, it allows task-specific policies to learn and discover what modules to share across different tasks. Since the soft combination process is differentiable, both policy network and the routing network can be trained together in an end-to-end manner. Note that the network for the soft Q-function follows the similar structure but initialized and trained independently.

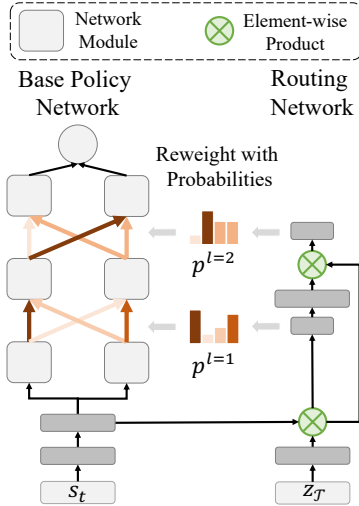

Figure 2: Our framework contains a base policy network with multiple modules (left) and a routing network (right) generating connections between modules in the base policy network.

Although the soft modularization provides a differentiable way to modularize and share the network across tasks, different tasks can still learn and converge with different training speed based on the difficulties of the tasks. For example, learning "reaching" policy is usually much faster than learning "pick and place" policy. To tackle this problem, we introduce a simple way to automatically adjust the losses for different tasks to balance the training across tasks.

In the following subsections, we will first introduce our network architecture with soft modularization, and then training objective for multi-task learning with this architecture.

### 4.1 Soft Modularization

As shown in Figure 2, our model of multi-task policy contains two networks: the base policy network and the routing network. At each time stage, the network takes the input of the current state $s_t$ and the task embedding $z_{\mathcal{T}}$ as inputs. We use an one-hot vector for $z_{\mathcal{T}}$ representing each task. We forward $s_t$ to a 2-layer MLP and obtain a $D$-dimension representation $f(s_t)$, which is then used as inputs for the modules as well as the routing network. We extract the representation for the task embedding by one fully connected layer as $h(z_{\mathcal{T}})$, which is also in $D$-dimension.

**Routing Network.** The depth of our routing network is corresponding to number of module layers in the base policy network. Supposed we have $L$ module layers and each layer has $n$ modules in the base policy network. The routing network will have $L - 1$ layers to output the probabilities to weight the modules and the dimension of the probability vector is $n \times n$. We define the output probability vector for the $l$th layer as $p^l \in \mathbb{R}^{n^2}$. The probability vector for the next layer can be represented as,

$$p^{l+1} = \mathcal{W}_d^l(\text{ReLU}(\mathcal{W}_u^l p^l \cdot (f(s_t) \cdot h(z_{\mathcal{T}})))), \tag{5}$$

where $\mathcal{W}_u^l$ is a fully connected layer in $\mathbb{R}^{D \times n^2}$ dimensions, which converts the probability vector to an embedding which has the same dimension as the task embedding representation and observation representation. We perform element-wise multiplication between these three embeddings to obtain a new feature representation, combining the information from the probabilities of previous layer, the observation and the task information. This feature is then forwarded to another fully connected layer $\mathcal{W}_d^l \in \mathbb{R}^{n^2 \times D}$, which leads to the probability vector for the next layer $p^{l+1}$. We visualize this process on computing $p^{l=2}$ from $p^{l=1}$ in Figure 2. To compute the first layer of probabilities, we use the inputs from both the task embedding and the state representation as,

$$p^{l=1} = \mathcal{W}_d^{l=1}(\text{ReLU}(f(s_t) \cdot h(z_{\mathcal{T}}))), \tag{6}$$

where $f(s_t)$ is the feature representation of the state with $D$ dimensions. To weight modules in the base policy network, we use softmax function to normalize $p^l$ as,

$$\hat{p}_{i,j}^l = \frac{\exp(p_{i,j}^l)}{\sum_{j=1}^{n} \exp(p_{i,j}^l)}, \tag{7}$$

which is the probability of weighting the $j$th module in the $l$th layer for contributing to the $i$th module in the $l + 1$ layer. We will illustrate how this is used in the base policy network in the following.

**Base Policy Network.** As shown in the left side of Figure 2, our base policy network has $L$ layers of modules, and each layer contains $n$ modules. We denote that the input for the $j$th module in the $l$th layer is a $d$-dimensional feature representation $g_j^l \in \mathbb{R}^d$. The input feature representation for the $i$th module in the $l + 1$ layer can be represented as,

$$g_i^{l+1} = \sum_{j=1}^{n} \hat{p}_{i,j}^l (\text{ReLU}(W_j^l g_j^l)), \tag{8}$$

where $W_j^l \in \mathbb{R}^{d \times d}$ represents the module parameters. We compute a weighted sum of the module outputs with the routing probability outputs. Recall from Eq. 7 that $\hat{p}_{i,j}^l$ represents the probability connecting the $j$th module in layer $l$ to the $i$th module in layer $l+1$ and it is normalized to $\sum_j \hat{p}_{i,j}^l = 1$. Given the final layer module outputs, we compute the mean and variance of the action as the outputs,

$$\mu, \sigma = \sum_{j=1}^{n} W_j^L g_j^L, \tag{9}$$

where $W_j^L \in \mathbb{R}^{d \times o}$ are the module parameters in the last layer, $o$ represents the output dimension.

Note that although we have only introduced the network architectures for policies so far, we adopt similar architectures with soft modularization for Q-function as well. The weights for both the base policy network and the routing network are not shared or reused in the Q-function.

### 4.2 Multi-task Optimization

We focus on the problem of balancing the learning across different tasks, as easier tasks usually converge faster than the harder ones. We scale the training objectives for the policy network with different weights for different tasks. These weights are learned automatically: the objective weight will be small if the confidence of the policy for the task is high, and be large if the confidence is low.

This loss weight is directly related to the temperature parameter $\alpha$ in SAC, trained via Eq. 2: When value of $\log \pi_\phi(a_t|s_t)$ become larger, which means entropy become smaller, $\alpha$ will become larger to encourage exploration. On the other hand, $\alpha$ will become small if $\log \pi_\phi(a_t|s_t)$ is small. We have different temperature parameters for $M$ different tasks: $\{\alpha_i\}_{i=1}^M$. The objective weights $w_i$ for task $i$ are proportional to the exponential of negative $\alpha_i$,

$$w_i = \frac{\exp(-\alpha_i)}{\sum_{j=1}^{M} \exp(-\alpha_j)}. \tag{10}$$

We then adjust the optimization objective from Eq. 3 as, $J_\pi(\phi) = \mathbb{E}_{\mathcal{T} \sim p(\mathcal{T})} [w_\mathcal{T} \cdot J_{\pi,\mathcal{T}}(\phi)]$, and the objetive for Q-fuction from Eq. 4 is adjusted as, $J_Q(\theta) = \mathbb{E}_{\mathcal{T} \sim p(\mathcal{T})} [w_\mathcal{T} \cdot J_{Q,\mathcal{T}}(\theta)]$.

## 5 Experiments

We perform experiments on multi-task robotics manipulation. We discuss the experiment environment, benchmark, and baselines, compare our method with baselines and conduct ablation study.

### 5.1 Environment

We evaluate our approach with the recent proposed Meta-World [43] environment. This environment contains 50 different robotics continuous control and manipulation tasks with a sawyer arm in the MuJoCo environment [38]. There are two challenges for multi-task learning in this environment: MT10 and MT50 challenge, which requires learning 10 and 50 manipulation tasks simultaneously. Building on top of these two challenges, we further extend the tasks to be goal-conditioned tasks. More specifically, the original MT10 and MT50 tasks are manipulation tasks with fixed goals. To make the tasks more realistic, we extend the tasks to have flexible goals. We name the two extensions as **MT10-Conditioned** and **MT50-Conditioned** tasks meanwhile we denote the original MT10 challenge as **MT10-Fixed** and the original MT50 challenge as **MT50-Fixed**.

### 5.2 Baselines and Experimental Settings

**Baselines.** We train our model with SAC [10]. We compare to five baselines with SAC without using our network architecture as following: (i) **Single-task SAC**: Individual policy for each task in MT10-Conditioned. (ii) **Multi-task SAC (MT-SAC)**: Using a one-hot task ID with the state as inputs. (iii) **Multi-task multi-head SAC (MT-MH-SAC)**: Built upon MT-SAC with independent heads for tasks. The same MT-SAC and MT-MH-SAC baselines are also proposed in [43], and we reproduce their results. (iv) **Mixture of Experts (Mix-Expert)**: It consists of four experts with the same architecture as MT-SAC, and a learned gating network for expert combination [13]. (v) **Hard Routing**: It consists of four module layers with four modules each layer. For each layer, agent selects one module to use according to the controller/router depending on the task, following [30].

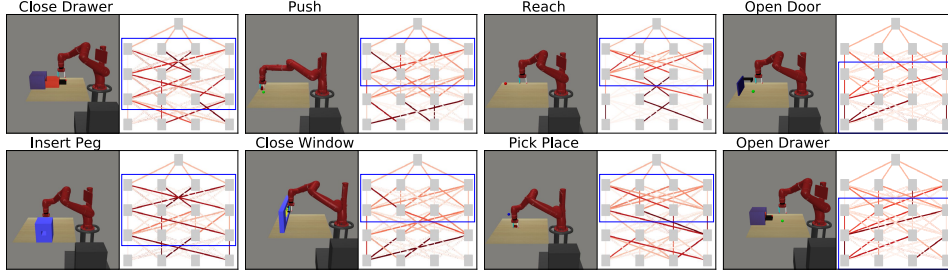

Figure 3: Sampled observation and corresponding routing. Each column shows two different tasks sharing similar routing. The shared parts are highlighted with blue boxes.

**Variants of Our Approach.** We conduct all experiments under two settings of our method. We ablate different numbers of module layer and modules in each layer: **Ours (Shallow)** contains $L = 2$ module layers, $n = 2$ modules per layer and each module outputs a $d = 256$ representation; **Ours (Deep)** contains $L = 4$ module layers, $n = 4$ modules per layer and each module outputs a $d = 128$ representation. The number of parameters is the same in both cases.

**Evaluation Metrics.** We evaluate the policies based on the success rate of executing the tasks, which is well-defined in the Meta-World environment[43]. We use the average success rate cross tasks to measure the performance. For each experiment, we train all methods with 3 random seeds. To plot the training curves, we plot the success rate of the polices across time with variance. For the final performance, we directly evaluate the final policy for each approach. We sample 100 episodes per task per seed. We compute the success rate for all these trials and report the averaged results.

**Training samples.** For MT-SAC and MT-MH-SAC baselines, we train them with 20 million samples on the MT10 setting and 100 million samples on the MT50 setting. For our method, Mix-Expert and Hard Routing Baselines, they converge much faster, and we train it with 15 million samples for MT10 and with 50 million samples for MT50 tasks.

## 5.3 Routing Network Visualization

We perform visualization on the networks trained with Ours (Deep) on the MT10-Conditioned setting.

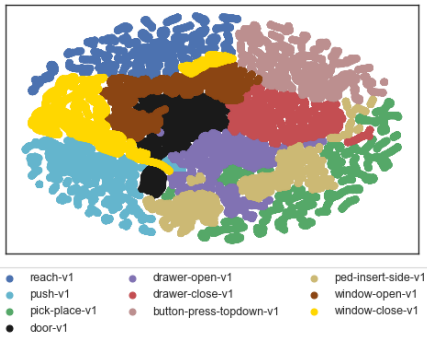

reach-v1   drawer-open-v1   ped-insert-side-v1
push-v1   drawer-close-v1   window-open-v1
pick-place-v1   button-press-topdown-v1   window-close-v1
door-v1

Figure 4: Probabilities from the routing network for different tasks are extracted and visualized with t-NSE. Routing probabilities from different tasks are grouped in different clusters.

**Probability Visualization.** We visualize the probabilities $p^l$ predicted by the routing network. Ours (Deep) contains $l = 4$ module layers with $n = 4$ modules per layer. As shown in Figure 3, we plot $p^l$ as the connections between different modules and use deep red color to represent large probability and light red color for small probability. For each column, we visualize the routing networks for two different tasks. We can see that even for different tasks, they could share similar module connections. It shows that our soft modularization method allows the reuse of skills across different tasks.

**t-SNE Visualization.** We visualize the routing probabilities for different tasks via t-SNE [39] in Figure 4. We run the policy on each task in MT10-Conditioned multiple times to collect routing samples. We combine all the routing probabilities from all layers into a $(l-1)n^2 = 48$ dimensional vector representing the routing path and visualize via t-SNE. We find clear boundaries between tasks, indicating that the agent can distinguish different tasks and choose corresponding skillset for each task. Besides, we notice that those tasks sharing similar task structures (e.g., drawer-open-v1 and drawer-close-v1, window-open-v1 and window-close-v1) are close in the plot.

## 5.4 Quantitative Results

**Results on MT10-Fixed.** As shown in Table 1, our re-implementation of multi-task multi-head SAC performs very close to the reported results in [43]. Although the final success rate of our method is only 2% better than our best baseline implementation, our method converges faster than the baselines, as shown in the 2nd plot in Figure 5. We are not getting a significant gain in the final success rate is because training 10 tasks with fixed goals is quite simple. We move forward to a more practical and challenging setting with training goal-conditioned policies.

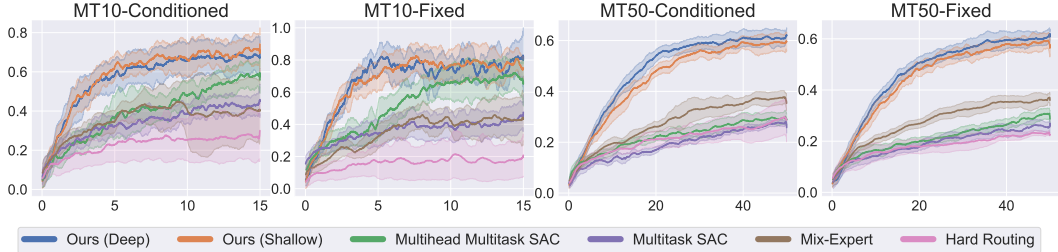

Figure 5: Training curves of different methods on all benchmarks (Concrete lines: the average over 3 seeds; Shaded areas: the standard deviation over 3 seeds). For MT10, our method converges much faster than the baselines. For MT50, we achieve a large gain on sample efficiency and performance.

| Method | MT10-Fixed | MT10-Conditioned | MT50-Fixed | MT50-Conditioned |
|---|---|---|---|---|
| MT-SAC* | 39.5% | - | 28.8% | - |
| MT-SAC | 44.0% | 42.6% | 31.4% | 28.3% |
| MT-MH-SAC* | **88.0%** | - | 35.9% | - |
| MT-MH-SAC | 85.0% | 67.4 | 35.5% | 34.2% |
| Mix-Expert | 42.8% | 40.0% | 36.1% | 37.5% |
| Hard Routing | 20.8% | 27.0% | 22.9% | 29.1% |
| Ours (Shallow) | 87.0% | **71.8%** | 59.5% | 60.4% |
| Ours (Deep) | 86.7% | 68.4% | **60.0%** | **61.0%** |

Table 1: Comparisons on average success rates for MT10 and MT50 tasks. MT-SAC*, MT-MH-SAC* indicate results reported in [43]. Approaches without * indicate baselines of our own implementation.

**Results on MT10-Conditioned.** As task difficulty increases, we can see from Table 1 that our approach (Ours (Shallow)) achieves more than 4% improvement over the baseline. Our approaches continue to improve the sample efficiency over the MT-MH-SAC baselines (1st plot in Figure 5).

**Results on MT50-Fixed and MT50-Conditioned.** When we are moving from joint training with 10 tasks to 50 tasks, the problem becomes more challenging. As shown in Table 1 and the last two plots in Figure 5, our method achieves a significant improvement over the baseline methods (around 24%) in both the fixed goal and goal-conditioned settings. We also observe that in MT50 environments, Ours (Deep) performs better than Ours (Shallow) approach, while it is the opposite in the MT10 setting. The reason for this phenomenon might be: (i) for a smaller number of task (MT10), simple network topology facilitates more on information sharing across tasks; (ii) for larger number of task (MT50), more complex network topology provides more routing choices and prevents different tasks from harming the performance of each other. It is also worthy of mentioning that our method achieves better success rates in MT50-Conditioned environments than MT50-Fixed. The reason is that MT50-Conditioned provides more examples in training for better generalization.

**Mixture of Experts and Hard Routing baselines.** We notice that although the performance of Mixture of Experts is only close to MT-SAC on MT10, it performs better than MT-MH-SAC on MT50. The reason is that when the number of tasks is small, Mixture of Experts is easy to degenerate to MT-SAC (with a single network). When the number of tasks becomes larger, the gating network in mixture of experts can learn to cluster the tasks into different sets corresponding to different experts. Similarly, while the Hard Routing baseline performs poorly in MT10, it catches up with the MT-SAC when applied to 50 tasks. It shows that routing still helps when task number increases. However, the optimization with hard routing is extremely challenging (see discussions in Section 2). Both baselines perform significantly worse than our method in both MT10 and MT50 tasks.

## 5.5 Effects on Network Capacity

We conduct experiments to see how the capacity of the network (number of parameters) can influence the performance of the baseline methods. We compare our approach with baselines using different numbers of parameters for MT50-Fixed in Table 2. We ablate different number of network layers and the number of hidden units in each layer. We denote MT-MH-SAC-$l$ as the multi-task multi-head SAC baseline with $l$ layers. We also ablate more hidden unites for each layer and name the methods with "Wide". The detailed configurations for different ablations are shown in Table 2.

We observe that even our model uses the smallest number of parameters, we can still achieve much better results. For example, our method is around 10% better than the baseline (MT-MH-SAC-5-Wide) which has 4.2x number of parameters compared to our method. We also observe that the gain

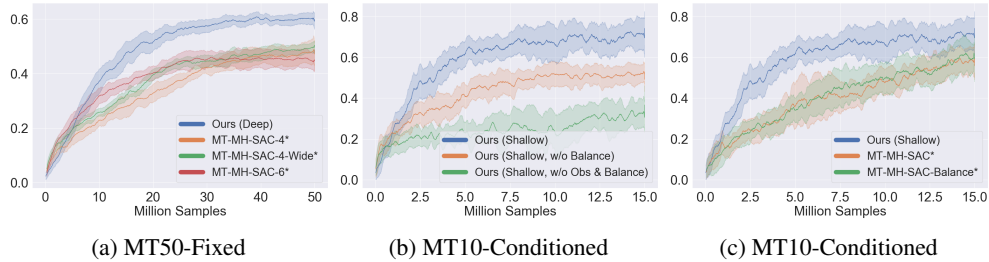

|                     | (a) MT50-Fixed | (b) MT10-Conditioned | (c) MT10-Conditioned |
| :-----------------: | :------------: | :------------------: | :------------------: |

Figure 6: (a) Compare Ours (Deep) and baselines with different network capacity for MT50-Fixed. (b) Analyse balancing training samples and using observation for routing network in Ours (Shallow) for MT10-Conditioned. (c) Analyse balancing training samples in the baseline for MT10-Conditioned.

| Method | MT50-Fixed | Params | layers | units |
| :--- | :---: | :---: | :---: | :---: |
| MT-MH-SAC* | 35.9% | 1.2x | 3 | 400 |
| MT-MH-SAC | 35.5% | 1.2x | 3 | 400 |
| MT-MH-SAC-4 | 46.7% | 1.6x | 4 | 400 |
| MT-MH-SAC-5 | 45.2% | 2.0x | 5 | 400 |
| MT-MH-SAC-6 | 45.0% | 2.4x | 6 | 400 |
| MT-MH-SAC-4-Wide* | 50.7% | 3.3x | 4 | 600 |
| MT-MH-SAC-5-Wide* | 50.3% | 4.2x | 5 | 600 |
| Ours (Deep) | **60.0%** | 1x | - | - |

Table 2: Comparison with baselines using different number of parameters for MT50-Fixed.

saturates very fast as we make the network larger and larger: The baseline with 4.2x capacity is slightly worse than the baseline with 3.3x capacity. We visualize the training curve in Figure 6a: our method converges faster and has much better performance than large capacity baselines.

## 5.6 Comparison with Single Task Policy

A substantial advantage of multi-task learning is with sample efficiency. We compare our policy with single task policy on MT10-Conditioned. Given 15 million samples, Ours (Shallow) achieve $71.8\%$ average success rate, while by average single task policy achieved $78.5\%$ success rate. Though the single task policy can overfit easily given enough training examples and achieve a very good result for one specific task but our method can still perform reasonably close to the single task policy, even we train with much fewer examples with much fewer parameters via a shared network. It shows that for each task, using data from other tasks along with our method can significantly improve sample efficiency, and skills learned by the soft modular policy can be shared between tasks with routing.

## 5.7 Analysing Learning Components

We analyze the importance of two learning components in our method with MT10-Conditioned: (i) Balance the training across different tasks using temperature parameters (Eq. 10); (ii) Use observation representation as the inputs for the routing network. We report the comparison results in Figure 6b and Figure 6c. In Figure 6c, we ablate our method in the Ours (Shallow) setting, and remove the balance training (Ours (Shallow, w/o Balance)) as well as remove both the balance training and observation inputs for the routing network (Ours (Shallow, w/o Obs & Balance)). If we remove one or both learning components, the success rate is reduced by a large margin. Thus both components play an important role in our approach. The importance of encoding observation for our routing network might also be a reason for the poor performance of Hard Routing baseline, since the controller of Hard Routing is parameterized by tabular lookup table which can not encode high dimensional information like observation. We also apply the balance training strategy to the baseline in Figure 6c as MT-MH-SAC-Balance. Interestingly, we find that the baseline approach is not affected as much with the new optimization strategy. Thus we do not apply balance training for baselines.

## 6 Conclusion

In this paper, we propose multi-task RL with soft modularization for robotics manipulation tasks. Our method improves the sample efficiency as well as the success rate over the baselines by a large margin. The advantage becomes more obvious when given more diverse tasks. This shows that soft modularization allows effective sharing and reusing network components across tasks, which opens up future opportunities to generalize the policy to unseen tasks in a zero-shot manner.

## 7 Potential Broader Impact

Our work provided a simple and effective framework for skill and component reuse in the multi-task RL domain, which the community can build off. With the learned skill module, Our work can also inspire work on zero-shot skill transferring and sharing.

With improved sample efficiency and potential zero-shot skill transfer, the community might be able to use reinforcement learning to solve tasks that not feasible before and build the robots that can generalize to different tasks. These robots could potentially bring lots of new possibilities in almost every aspect of people's daily life, e.g., self-driving cars and house-hold robots. Besides, general learned robotics can also be useful for unseen or urgent out-of-distribution scenes. For instance, when it comes to performing a rescue under the earthquake, the robot should have the ability to cope with different conditions.

In the deep learning era, collecting samples and training large models could consume a lot energy and release a massive amount of carbon dioxide. With better sample efficiency, training reinforcement learning policy for real-world settings can be much more environment-friendly. Meanwhile, better sample efficiency can also lower the bar and be more accessible for inexperienced researchers to get into the field.

**Acknowledgement**: This work is supported, in part, by grants from DARPA LwLL, NSF 1730158 CI-New: Cognitive Hardware and Software Ecosystem Community Infrastructure (CHASE-CI), NSF ACI-1541349 CC*DNI Pacific Research Platform, a research grant from Qualcomm, and gift from TuSimple.

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
