[Supplementary Material]

# Multi-Task Reinforcement Learning with Soft Modularization (Supplementary Material)

## 1 Hyperparameters

In this section, we provide the detailed hyperparameter value for each methods in our experiment. The hyperparameters for single task sac is chose to align with our method on MT10-Condiitioned.

### 1.1 Single Task SAC

| Hyperparameter | Hyperparameter values |
|---|---|
| network architecture | feedforward network |
| network size | three fully connected layers with 400 units |
| batch size | 1280 |
| non-linearity | ReLU |
| policy initialization | standard Gaussian |
| # of samples / # of train steps per iteration | 10 env step / 1 training step |
| policy learning rate | 3e-4 |
| Q function learning rate | 3e-4 |
| optimizer | Adam |
| discount | .99 |
| horizon | 150 |
| reward scale | 1.0 |
| temperature | learned |

### 1.2 Multi-Task SAC

| Hyperparameter | Hyperparameter values |
|---|---|
| network architecture | feedforward network |
| network size | three fully connected layers with 400 units |
| batch size | $128 \times$ number_of_tasks |
| non-linearity | ReLU |
| # of samples / # of train steps per iteration | number_of_tasks env steps / 1 training step |
| policy learning rate | 3e-4 |
| Q function learning rate | 3e-4 |
| optimizer | Adam |
| discount | .99 |
| horizon | 150 |
| reward scale | 1.0 |
| temperature | learned and disentangled with tasks |

### 1.3 Multi-Task Multi-Headed SAC

| Hyperparameter | Hyperparameter values |
|---|---|
| network architecture | multi-head (one head per task) |
| network size | three fully connected layers with 400 units |
| batch size | $128 \times$ number_of_tasks |
| non-linearity | ReLU |
| # of samples / # of train steps per iteration | number_of_tasks env steps / 1 training step |
| policy learning rate | 3e-4 |
| Q function learning rate | 3e-4 |
| optimizer | Adam |
| discount | .99 |
| horizon | 150 |
| reward scale | 1.0 |
| temperature | learned and disentangled with tasks |

## 1.4 Modular Policy-Ours Shallow

| Hyperparameter | Hyperparameter values |
|---|---|
| number of module layers | 2 |
| number of modules per layer | 2 |
| number of module hidden units | 256 |
| representation size | 400 |
| batch size | $128 \times$ number_of_tasks |
| non-linearity | ReLU |
| # of samples / # of train steps per iteration | number_of_tasks env steps / 1 training step |
| policy learning rate | 3e-4 |
| Q function learning rate | 3e-4 |
| optimizer | Adam |
| discount | .99 |
| horizon | 150 |
| reward scale | 1.0 |
| temperature | learned and disentangled with tasks |

## 1.5 Modular Policy-Ours Deep

| Hyperparameter | Hyperparameter values |
|---|---|
| number of module layers | 4 |
| number of modules per layer | 4 |
| number of module hidden units | 128 |
| representation size | 400 |
| batch size | $128 \times$ number_of_tasks |
| non-linearity | ReLU |
| # of samples / # of train steps per iteration | number_of_tasks env steps / 1 training step |
| policy learning rate | 3e-4 |
| Q function learning rate | 3e-4 |
| optimizer | Adam |
| discount | .99 |
| horizon | 150 |
| reward scale | 1.0 |
| temperature | learned and disentangled with tasks |