[Reviews · NeurIPS 2020]

Review 1

Summary and Contributions: In this article, the authors present a new method in the field of multi-task Reinforcement Learning. While the method is not restricted to a certain domain, they investigate the method in the experimental part in the application domain of manipulation, using an existing manipulation task benchmark suite (Meta-World). The main issues with multi-task RL that the authors motivate in the introduction and use to motivate their method are: conflicting gradients and balancing optimisation between tasks. They address important issues in multi-task RL that typically hurt the performance gain that we expect in terms of data efficiency and final performance, reported in all major publications in the field. From a high level perspective, there are two main ideas in the paper. The first is to develop an alternative network topology / structure for the policy (and the Q function). They use an explicit modularization technique by splitting the network in a base and a routing network. The modular topology is still backward differentiable and both networks can in principle be trained with a default RL algorithm. The second is to use the Soft Actor Critique (SAC) algorithm, introduce weights for the task losses and connect them to the learned temperature alpha in SAC, so that they are balanced during learning. In the experiment section they show that their proposed method can outperform a set of baselines and does this by a large margin for the more difficult multi-task setting.

Strengths: The authors motivate their work very well and embed it sufficiently in the scientific context. As mentioned above, they address an important and highly relevant topic in the field of multi-task RL. The description of the network topology, the routing and the connection to SAC is technical sound and sufficient to understand and follow the main parts of the work. The experimental results look technical sound and the ablations and results are sufficient to support their method and to show the advantages for multi-task RL.

Weaknesses: While a lot of parameters are given and make the paper and results strong, the Q function is treated a bit differently. The authors only mention to use a “similar” topology compared to the policy. Obviously this leaves some room for speculation as the action has to be connected to the network, it is not clear how the topology would look exactly and the reader can not be sure if the number of weights etc. in the Q function is known for reproducibility. Please add a few more details about the Q-function. I wondered a bit why the aspect of not only sharing weights, but also sharing data was not discussed. This would make the paper even stronger in my opinion.

Correctness: All results, the description and derivation of the method, as well as the empirical methodology look correct and technical sound.

Clarity: I liked to read the paper, as it is well written, uses a clear language, is well structured and is good to understand. I liked to read the paper, as it is well written, uses a clear language, is well structured and is good to understand. The only point where I struggled a bit was reading and understanding the detailed description of the policy topology. While all information seems to be there it is still a bit cumbersome to get all the details and to understand why some parts are there and what they are doing exactly.

Relation to Prior Work: The authors embed their work in the most important literature and other approaches in the field. As mentioned already above the only thing that could be added is to discuss different methods that not only share weights but also share data from multiple tasks (as done e.g. in 27).

Reproducibility: Yes

Additional Feedback: As already mentioned I liked to read the paper. What I was wondering is what influence the Q function - and possibly different architectures, topologies, sizes - would have on the results. While this is not strictly in the scope of the paper some ablations could make sense. As mentioned above, at least a few more details about the function should be added. As also mentioned I wondered if sharing data (e.g. relabeling the reward) would help.


Review 2

Summary and Contributions: This manuscript presents a method for multi-task learning based on a novel neural architecture. The architecture is composed of two elements: a neural network with encapsulated units at multiple levels and a second network that estimates the strength of the connectivity between each unit at one level and the units of the next level. This creates a meta-network level. The authors apply this neural architecture to an actor-critic policy learning architecture (SAC) for reinforcement learning, both in the critic and the actor, and demonstrate that their approach performs better (generalization among tasks) than simple baselines and ablated versions on robot simulated manipulation tasks. I updated my review positively after the rebuttal.

Strengths: -Relevant problem: meta-learning is attracting more attention since it is clear than training a separate policy for each task is not a feasible approach for robotics. Finding the right way to share information between tasks is an open research question. -Novelty: the idea seems to be new, although related to other concepts like capsule networks -Results: compared to the baselines, the results of the proposed method are better

Weaknesses: -Theoretical grounding: the paper is not well grounded in neural network theory. It is unclear what is the effect of different architectures for the units. Why a dot product for the weighting? -Experimental evaluation: there have been multiple methods from the community for multi-task learning, many of them included in the related work and with available source code, but none of them are included in the evaluation. At least, a comparison to state-of-the-art or a good representative (e.g. MAML). -Writing: the paper is not well written, with multiple typos, missing articles, errors in capitalization and strange sentences that make it hard to read.

Correctness: The method seems correct.

Clarity: No, the text could improve

Relation to Prior Work: The related work section is rather superficial. The idea of hierarchical architectures (a networks that outputs the architecture for another) is old and there are many examples in reinforcement learning and other AI fields (e.g. computer vision) that they authors should compare to, for example, capsule networks.

Reproducibility: Yes

Additional Feedback: These elements should improve to make the manuscript and contribution clearer: - Improve text - Add experiments comparing to any of the state of the art multi-task learning methods - Improve the theoretical grounding of your design decisions


Review 3

Summary and Contributions: The paper presents an approach for alleviating multi-task interference when training robotic policies with reinforcement learning. The paper introduces a routing network conditioned on the task ID and state that outputs probabilities for each module of a base policy network that determines its weight when executing the policy. Each module has a range of hidden units, and the policy network consists of multiple layers of such modules. The paper uses Soft Actor-Critic (SAC) to jointly train the policy and routing networks. The approach is evaluated on Meta-World tasks and it is shown to outperform previous SAC-based multi-task methods especially when adding goal conditioning to the tasks.

Strengths: - Multi-task learning is a very important problem for scaling up reinforcement learning and providing a solution for multi-task interference is crucial to successfully employ multi-task learning. - Soft combination of the network modules leads to a better structure sharing between tasks than hard routing. - Presented approach is shown to outperform previous multi-task routing strategies on up to 50 tasks, especially in a challenging goal-conditioned setup.

Weaknesses: - It would be interesting to see an evaluation on higher-dimensional input spaces, such as camera images. - It would be interesting to see a discussion on how many layers of the base policy should be “routed” for a successful application of the method. In the current framework, two layers that process the state are not influenced by the routing network. However, as we start to use higher-dimensional spaces such as images, we might want to skip routing for larger parts of the network.

Correctness: Claims and mathematical derivations in the paper are coherent and empirical evaluation correctly shows performance improvements over the baselines.

Clarity: The paper is well-written and easy to understand and follow.

Relation to Prior Work: The paper establishes connection to prior works and uses them as baselines for the experiments. It would be interesting to see a discussion how the proposed method connects to the FiLM-approach, originally proposed for conditioning in visual reasoning [1]. [1] FiLM: Visual Reasoning with a General Conditioning Layer. E. Perez, F. Strub, H. de Vries, V. Dumoulin, A. Courville. 2017.

Reproducibility: Yes

Additional Feedback: Post-rebuttal comments: Thanks for addressing my comments and providing new details in the rebuttal. Please incorporate these details in the final version of the paper.


Review 4

Summary and Contributions: The paper considers the optimization problem of multi-task reinforcement learning. It addresses what parameters in the network should be reused across tasks, and how the gradients from different tasks may interfere with each other. The paper introduces an explicit modularization technique on policy representation and a routing network that is soft modularization to softly combine all the possible routes. The paper considers different sub-modules and a routing network that compose multiple modules. In the paper the soft combination process is differentiable.

Strengths: The problem addressed by the paper is very important. The idea using different sub-modules and a routing network that compose multiple modules is also interesting. The experimental results are very good.

Weaknesses: The idea is interesting. Learning modules and routing networks is very interesting. However, the idea is not new. Previous works (Rosenbaum et al.) have the similar idea on using routing networks. The paper argues that the previous routing policy suffers from exponentially higher variance in policy gradient. It is better to have analysis about the variance between different methods. In Figure 5, for MT10-Fixed and MT50-Fixed, the new method and Hard Routing have similar variance. In Figure 6, there is no improvement on the variance. Another concern is why just learn masks or weights directly for the base policy network. What is the difference between using masks or weights on modules and the routing network? A solution may use gated networks for these sub-modules. Is the work related to NAS (Neural Architecture Search) ? How to learn the parameters of the routing network?

Correctness: Correct.

Clarity: The paper is written well.

Relation to Prior Work: Clear.

Reproducibility: Yes

Additional Feedback: Update after the response: I read the response and other reviews. The response does not answer my concerns. So I do not change the score.

[Author Response · NeurIPS 2020]

In this paper, we propose to tackle the problem of multi-task reinforcement learning with a novel modular network model. Instead of using hard selection on modules, we introduce a method called *soft modularization* which softly combines the modules. Our approach enables efficient optimization and sharing across modules. The role of each module can automatically emerge after training without manual specification. We perform extensive **empirical** studies and show significant improvement (**>20%** success rate) over state-of-the-art approaches in the robot manipulation tasks (50 multi-task). We are glad to receive positive feedbacks on our work for both the novelty and the performance: R1:"motivate their work very well, is technical sound," R2:"idea seems to be new," R3:"very important problem, better structure sharing," R4: "the experimental results are very good." We will address reviewers' comments as follows.

———————————————————————————— **For Reviewer #2** ————————————————————————————

**Theoretical grounding: the paper is not well grounded in neural network theory.** While theory is important, our work is focusing on designing novel a modular network for multi-task RL and **empirically** showing its advantage. We are confused about what "neural network theory" R2 is asking for. R2 also asks "Why a dot product for the weighting?" But weighting itself indicates multiplication. R2 has not provided an alternative way for weighting.

**Meta-learning is attracting, Comparison to state-of-the-art (e.g. MAML).** R2 may have misunderstood that our work with meta-learning. We emphasize here that this paper is tackling **multi-task** learning but not meta-learning. Thus meta-learning approaches like MAML do not apply. We also stress that we have compared to all the baselines we can find codes for and implement including Mixture of Experts [12], Hard Routing [29], and Muti-task Muti-head SAC [43], which is the previous state-of-the-art. R2 also has not provided a reference on multi-task RL for us to compare.

**Comparing to hierarchical architectures like capsule networks.** Capsule networks have not been applied to multi-task RL. How to adopt it in multi-task RL is an interesting direction to study, but it is out of the scope of our paper.

**Writing.** While R2 complains about our writing, other reviewers all have positive feedback: "I liked to read the paper, as it is well written"(R1), "The paper is well-written and easy to understand"(R3), "The paper is written well."(R4).

———————————————————————————— **For Reviewer #4** ————————————————————————————

**Similar idea with Rosenbaum et al.** We have extensively addressed the difference between our work and Rosenbaum et al. in both related work (line 83-90) and compare against it in the experiment (denoted as Hard Routing [29]). Our soft modularization approach improves over it in both sample efficiency and performance significantly.

**Similar Variance for Hard Routing and Ours.** Suffering from higher variance in policy gradient, the success rate (22.9% for MT50) of Hard Routing is significantly lower than our approach (60.0% for MT50). Although the result of Hard Routing has similar variance as our method, this only means all their results are equally bad. In an extreme case, multiple random policies will have 0% accuracy but zero variance. Thus higher variance in gradient does not necessarily convert to higher variance in performance.

**Gating/Masking modules instead of routing.** We can see gating modules as a special case of routing mechanism, where all the routes connected to the same module will be weighted by the same scale parameter. Our routing network in the paper allows the routes connected to the same module be weighted differently, leading to better flexibility.

**Training of routing network.** As we mentioned in our paper (line 131-132), the soft combination method we proposed is fully differentiable, so both our base policy network and routing network can be trained together end-to-end.

———————————————————————————— **For Reviewer #1** ————————————————————————————

**Modular Structure for Q function.** For the base network, we concat state and action then feed them as inputs, and output the value. For the routing network, the inputs are the same as the policy including both states and task embedding.

**Sharing Data.** Sharing data will be an interesting future direction, and it is complementary to our current approach.

———————————————————————————— **For Reviewer #3** ————————————————————————————

**High dimensional inputs like images.** While learning with image inputs is an exciting direction, it is unclear how to learn a good visual representation for RL, which is a common challenge for vision-based RL. Recent solutions involve self-supervised visual representation learning, which introduces extra complexity. We will study this in the future.

**How many layers should be routed?** We study this problem in two directions: (i) Increasing the routing layers: we have compared our model with 2 routing layers (Ours (Shallow)) and 4 routing layers (Ours (Deep)) in our experiments, and we find improvement by using 4 layers in MT50 tasks. (ii) Increasing the layers before routing starts: We experiment with different number of layers of FC (2,3,4 layers) before routing starts and do not observe obvious performance difference. The reason might be the current input states are in low-dimension. For high dimension visual inputs, we hypothesize that we can first use ConvNets to extract the visual representation in lower-dimension, and then apply our routing modular networks on top of the extracted representation.

**Comparing to FiLM (Perez et.al).** This works predicts input conditioned feature, and our work predicts task conditioned network routing. While related, they are also tackling very different tasks. We will include and discuss this paper in our related work.

[Meta-Review · NeurIPS 2020]

The reviewers agreed that this is a reasonably well-written paper, on an important topic, with excellent empirical results. Given that the level of enthusiasm varied widely across reviewers, I'd recommend revising the final paper for more clarity, especially with respect to the novelty of the ideas.